# RIEMANNIAN GENERATIVE DECODER

## ABSTRACT

Riemannian representation learning typically relies on an encoder to estimate densities on chosen manifolds. This involves optimizing numerically brittle objectives, potentially harming model training and quality. To completely circumvent this issue, we introduce the *Riemannian generative decoder*, a unifying approach for finding manifold-valued latents on *any* Riemannian manifold. Latents are learned with a Riemannian optimizer while jointly training a decoder network. By discarding the encoder, we vastly simplify the manifold constraint compared to current approaches which often only handle few specific manifolds. We validate our approach on three case studies — a synthetic branching diffusion process, human migrations inferred from mitochondrial DNA, and cells undergoing a cell division cycle — each showing that learned representations respect the prescribed geometry and capture intrinsic non-Euclidean structure. Our method requires only a decoder, is compatible with existing architectures, and yields interpretable latent spaces aligned with data geometry. A temporarily anonymized codebase is available on: https://anonymous.4open.science/r/rgd-470F.

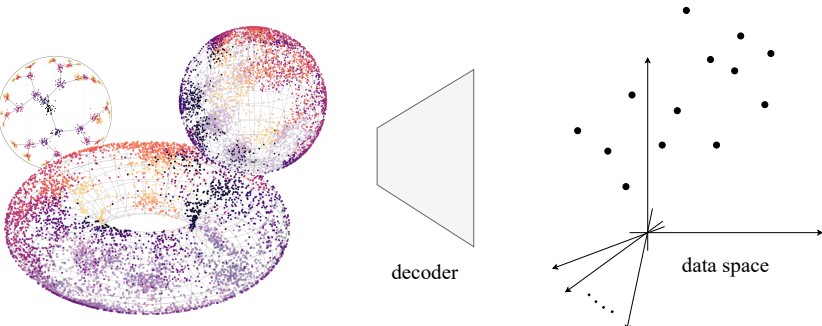

Figure 1: Our decoder reconstructs data from Riemannian manifolds where representations are learned as model parameters via maximum a posteriori.

## 1 INTRODUCTION

Real-world data often lie on non-Euclidean manifolds — e.g., evolutionary trees, social-network graphs, or periodic signals — yet most latent-variable models assume $\mathbb{R}^d$ latent spaces. Euclidean methods often fail to provide visualizations rooted in the geometry we know underlies the data, completely missing clear signals (Section 4.3). Meanwhile, low-dimensional projections *directly guide* how practitioners interpret their data in various fields. While non-linear projections like UMAP are greatly used and abused (Huang et al., 2022), having more control of the projection facilitates better hypothesis-based exploration of data. For this, *Riemannian manifolds* — spaces that are locally Euclidean but endowed with a smoothly varying inner product (metric) defining lengths, angles, geodesics, and curvature — provide a general framework for modeling geometry. Existing works have adjusted variational autoencoders (VAEs) for embedding data onto various geometries. However, despite the flexibility of VAEs, enforcing manifold priors (e.g., von Mises–Fisher on spheres or Riemannian normals in hyperbolic spaces) requires complex densities and Monte Carlo estimates of normalizing constants, limiting scalability for general manifolds.

We therefore propose the *Riemannian generative decoder*: we discard the encoder and directly learn manifold-valued latents with a Riemannian optimizer while training a decoder network. This encoderless scheme removes the need for approximate densities on the manifold, and handles any Riemannian manifold — including products of heterogeneous manifolds. With a *geometry-aware regularization* through input noise, our model is further encouraged to penalize sharpness relative to the local curvature. We analyze this form of regularization and see its importance in preserving geometric structure during dimensionality reduction. Our contributions are as follows,

- We introduce **a unifying framework for representation learning** on *any* Riemannian manifold via combining Riemannian optimization with an encoder-less generative model,
- We introduce a **highly scalable *geometric regularization***, promoting coherency between a decoder function and a chosen manifold's metric through noise perturbation,
- We explore various **real-world biological datasets** and find our approach to match or improve a diverse set of metrics; all while being *much stabler* in high dimensions where other methods fail.

## 2 BACKGROUND

Learned representations often reveal the driving patterns of the data-generating phenomenon. Much of computational biology — and especially data-driven fields like transcriptomics — greatly rely on dimensionality reduction techniques to understand the underlying factors of their experiments (Becht et al., 2019). Unfortunately, a lack of statistical identifiability implies that such representations need not be unique (Locatello et al., 2019). Therefore, it is common practice to inject various inductive biases that reflect prior beliefs or hypotheses about the analyzed problem. One way is to impose a specific geometry on the latent space.

### 2.1 LATENT VARIABLE MODELS

Autoencoders (AEs) learn a deterministic mapping $x \mapsto z \mapsto \hat{x}$ by minimizing a reconstruction loss

$$\min_{\theta,\phi} \sum_{i=1}^{N} L\left(x_i, f_\theta(g_\phi(x_i))\right) \tag{1}$$

where $x_1, \ldots, x_N$ are the training samples, $L$ is the loss function, e.g. the squared error, $g_\phi$ is the encoder and $f_\theta$ the decoder. Because $f_\theta$ is typically smooth, nearby latent codes produce similar reconstructions. This imposes a *smoothness bias* on the representation: distances in latent space are tied to distances in data space.

The variational autoencoder (VAE) by Kingma et al. (2013) extends this by introducing a prior $p(z)$, a stochastic encoder $q_\phi(z \,|\, x)$ as a variational distribution, and a stochastic decoder $p_\theta(x \,|\, z)$. The marginal likelihood

$$p(x|\theta) = \int p(x|z,\theta)p(z)dz \tag{2}$$

is intractable, but is lower bounded by the evidence lower bound (ELBO):

$$\log p(x|\theta) \geq \underbrace{\mathbb{E}_{q_\phi(z|x)}\left[\log p_\theta(x \mid z)\right]}_{\text{data reconstruction}} - \underbrace{D_{\mathrm{KL}}\left(q_\phi(z \mid x) \,\|\, p(z)\right)}_{\text{latent regularization}}, \tag{3}$$

where $D_{\mathrm{KL}}$ is the Kullback-Leibler divergence. The decoder is trained by maximizing the ELBO to reconstruct $x$ from samples of $z \sim q_\phi(z \mid x)$. The KL term encourages the encoding distribution to match the prior, typically $\mathcal{N}(0, I)$, while the stochasticity of $q_\phi$ forces the decoder to be robust to perturbations in $z$. Together, these constraints strengthen the smoothness bias across the encoder distribution.

An alternative to the VAE is the Deep Generative Decoder (DGD; Schuster & Krogh 2023), avoiding an encoder entirely. Each latent $z_i$ is treated as a free parameter, and the model uses MAP estimation by maximizing $P(z, \theta, \phi|x)$, corresponding to maximizing the following in $z$, $\theta$ and $\phi$:

$$(\hat{z}, \hat{\theta}, \hat{\phi}) = \arg\max_{z,\theta,\phi} \sum_{i=1}^{N} \left( \log p_\theta(x_i \mid z_i) + \log p(z_i \mid \phi) \right) + \log\left(P(\theta)P(\phi)\right) \tag{4}$$

The last term contains priors on $\theta$ and $\phi$. A parameterized distribution $p(z_i \mid \phi)$ on latent space, such as a Gaussian mixture model, can introduce inductive bias. The decoder smoothness imposes again a continuity constraint on $z \mapsto x$, as reconstructions must interpolate well across learned codes. Unlike the VAE, no amortized inference is used, but the same decoder regularization implicitly shapes the latent geometry.

In all three frameworks — AE, VAE, and DGD — the smoothness of the decoder function acts as a regularizer on latent codes. Since nearby $z$ produce similar outputs, the learned representations *inherit* geometric continuity. The VAE further strengthens this bias through stochastic encodings and KL regularization. The DGD enforces it by directly optimizing per-sample codes under a smooth decoder. These smoothness priors play a central role in learning meaningful low-dimensional structure.

## 2.2 GEOMETRIC INDUCTIVE BIASES

Most learned representations are assumed to be Euclidean, belonging to $\mathbb{R}^d$. This implies a simple, unbounded topological structure for the representations. This is a flexible and not very informative inductive bias. We briefly survey parts of the literature and generally find that existing approaches involve layers of complexity that potentially limit their performance.

**Spherical representation spaces** encode compactness and periodicity. Davidson et al. (2018) and Xu & Durrett (2018) define latents on $\mathbb{S}^{d-1}$ via a von Mises–Fisher prior:

$$p(z \mid \mu, \kappa) = C_d(\kappa) \exp(\kappa \mu^\top z) \quad \text{with} \quad C_d(\kappa) = \frac{\kappa^{d/2-1}}{(2\pi)^{d/2} I_{d/2-1}(\kappa)}, \tag{5}$$

where $\mu \in \mathbb{S}^{d-1}$ and $\kappa > 0$. Sampling uses rejection or implicit reparameterization and KL terms involve Bessel functions, complicating Equation 3 while adding computational overhead and bias.

**Hyperbolic representation spaces** effectively capture hierarchical data structures (Krioukov et al., 2010). A popular choice (used for, e.g., the $\mathcal{P}$-VAE (Mathieu et al., 2019)) is the Poincaré ball $\mathbb{B}^d$, with metric $g_z = \lambda(z)^2 I$, where $\lambda(z) = 2/(1 - \|z\|^2)$, and distance

$$d_{\mathbb{B}}(u, v) = \text{arcosh}\Big(1 + 2\frac{\|u - v\|^2}{(1 - \|u\|^2)(1 - \|v\|^2)}\Big). \tag{6}$$

One typically uses the Riemannian normal prior

$$p(z) \propto \exp\big(-d_{\mathbb{B}}(z, \mu)^2/(2\sigma^2)\big). \tag{7}$$

The ELBO then requires approximating both the intractable normalizing constant of the prior and volume corrections, typically via Monte Carlo or series-expansion methods. Alternative hyperbolic embeddings like the Lorentz (Nickel & Kiela, 2018) or stereographic projections (Skopek et al., 2019) improve computational stability and flexibility but face analogous challenges.

**General geometries** can represent different inductive biases (Kalatzis et al., 2020; Connor et al., 2021; Falorsi et al., 2018; Grattarola et al., 2019). Current literature is based on encoders whose densities generally lack closed-form formulas on arbitrary manifolds $\mathcal{M}$. They rely on approximations like Monte Carlo importance sampling, truncated wrapped normals

$$q(z|\mu, \Sigma) \approx \sum_{k \in \mathbb{Z}^d} \frac{\exp(-\frac{1}{2}\|\text{Log}_\mu(z) + 2\pi k\|^2_{\Sigma^{-1}})}{(2\pi)^{d/2}|\Sigma|^{1/2}}, \tag{8}$$

or random-walk reparameterization encoders such as $\Delta$VAE (Rey et al., 2019), that simulates Brownian motion using the manifold exponential map: $z = \text{Exp}_\mu(\sum_i \xi_i)$, $\xi_i \sim \mathcal{N}(0, \frac{t}{\text{steps}} I)$.

**Curvature regularization.** Independent of encoder–decoder choices, Lee & Park (2023) propose adding explicit intrinsic and extrinsic curvature penalties of the learned manifold. They derive regularizers that depend on second-order derivatives of the decoder — e.g., for intrinsic curvature:

$$\begin{aligned}
\text{IC}_{\text{approx}}(z) = \Big( &\tfrac{1}{2}(w{\cdot}\nabla)\big(w^\top G_f^{-2}(v{\cdot}\nabla)(G_f v)\big) - \tfrac{1}{2}(v{\cdot}\nabla)\big(w^\top G_f^{-2}(v{\cdot}\nabla)(G_f w)\big) \\
&+ \tfrac{1}{4} w^\top G_f^{-3}(v{\cdot}\nabla G_f)(v{\cdot}\nabla)(G_f w) - \tfrac{1}{4} w^\top G_f^{-2}(v{\cdot}\nabla G_f) G_f^{-1}(v{\cdot}\nabla)(G_f w) \\
&- \tfrac{1}{4} w^\top G_f^{-2}(v{\cdot}\nabla G_f) G_f^{-1}(w{\cdot}\nabla)(G_f v) + \tfrac{1}{4} w^\top G_f^{-1}(v{\cdot}\nabla G_f) G_f^{-2}(w{\cdot}\nabla)(G_f v)\Big)^2.
\end{aligned} \tag{9}$$

where $v, w \sim \mathcal{N}(0, I)$, $G_f = J_f(z)^\top J_f(z)$, and $(a \cdot \nabla)$ is a directional derivative. Computationally challenging second-order terms enter via $(a \cdot \nabla) G_f$ since $\partial J_f$ are Hessian–vector products of $f$. Their objective encourages globally "flat" embeddings in a Riemannian sense; in contrast, our geometry-aware noise induces a first-order Jacobian penalty which aligns local decoder smoothness with the chosen geometry while avoiding challenging computations (Section 3).

## 3 METHODOLOGY

Much of the difficulty in probabilistically learning representations over non-trivial geometries is that densities are notably more difficult to handle than, e.g., a Gaussian distribution. Inspired by Schuster & Krogh (2023) we propose not to variationally infer the representations, but instead perform standard maximum a posteriori estimation (MAP). Combining Riemannian optimization with a decoder and a form of geometry-aware regularization, we build a simple yet effective representation learning scheme that works across different geometric inductive biases.

### 3.1 MODEL FORMULATION

We generalize the DGD framework (Schuster & Krogh, 2023) to work with any latent geometry. Let $x \in \mathcal{X}$ denote observed data samples, and let $z \in \mathcal{M}$ represent latent variables constrained to a Riemannian manifold $\mathcal{M}$. We define a decoder function $f_\theta : \mathcal{M} \to \mathcal{X}$, parameterized by $\theta$, which maps latent representations to the data space. With a reconstruction metric, we perform MAP estimation of the decoder parameters $\theta$ and the latent representations $z = \{z_1, z_2, \ldots, z_N\}$ for a dataset $X = \{x_1, x_2, \ldots, x_N\}$. Equation 4 then leads to the training objective

$$(\hat{z}, \hat{\theta}) = \arg\max_{z, \theta} \sum_{i=1}^{N} (\log p(x_i \mid z_i, \theta) + \log p(z_i)) \tag{10}$$

where the prior on $\theta$ has been left out; one often uses weight decay instead. For compact manifolds, we assume a uniform distribution $\log p(z_i) = -\log \text{Vol}(\mathcal{M})$ and $\log p(z_i)$ need not be included in the loss. At generation time, one samples $z \sim \text{Uniform}(\mathcal{M})$ to compute $x = f_\theta(z)$. For non-compact manifolds, choices like the wrapped normal can function as $p(z)$. Note, however, that the generative properties are not the focus here.

Concretely, we directly *assign* a randomly initialized latent representation to each data sample. To enforce the manifold constraint, we use `RiemannianAdam` (Bécigneul & Ganea, 2018) on these latent representations. Such optimizers work by following the Riemannian gradient projected onto the tangent space and mapping the updated point back onto the manifold. Relying on *geoopt* (Kochurov et al., 2020) for defining tensors and optimizing on manifolds, the implementation becomes exceedingly simple (see Appendix A). One optimizer thus learns these points on the manifold (`RiemannianAdam`), while another learns the decoder parameters (`Adam`). The quality of each parameter set naturally affects the other, but training remains stable. Optimization during validation and test time is necessary as no encoder is available — unless one is trained post hoc — and follows the strategy of Schuster & Krogh (2023), freezing decoder parameters.

For manifolds whose metric tensor varies with position, we introduce a *geometry-aware regularization* to inform the model about the metric. During training, each latent $z$ is perturbed with Gaussian noise whose covariance is the *chosen manifold's* inverse Riemannian metric at $z$. This adapts the noise to local curvature: on homogeneous manifolds such as the hypersphere (where curvature is constant and metric variation merely reflects coordinate scaling) the procedure recovers nearly isotropic noise, whereas on spaces with non-uniform curvature the noise shape is greatly adjusted by location. We outline a derivation inspired by Bishop (1995) and An (1996) to analyze this noise:

Let $\epsilon \sim \mathcal{N}(0, \sigma^2 G^{-1}(z))$ and define the squared-error loss $L(z) = \|f(z, \theta) - y\|^2$ for some target $y$. We inject noise via the exponential map, which we approximate by the identity to $O(\|\epsilon\|^2)$:

$$z' = \text{Exp}_z(\epsilon) = z + \epsilon + O(\|\epsilon\|^2). \tag{11}$$

Ignoring higher order terms $o(\|\epsilon\|^2)$, a second-order Taylor expansion around $z$ gives

$$L(z') \approx L(z) + \nabla_z L(z)^\top \epsilon + \tfrac{1}{2} \epsilon^\top \nabla_z^2 L(z) \, \epsilon, \tag{12}$$

Taking expectation over $\epsilon$ and using $\mathbb{E}[\epsilon] = 0$, $\mathbb{E}[\epsilon\epsilon^\top] = \sigma^2 G^{-1}(z)$, we obtain

$$\mathbb{E}_\epsilon[L(z')] = L(z) + \frac{\sigma^2}{2} \operatorname{Tr}\big(\nabla_z^2 L(z) \, G^{-1}(z)\big). \tag{13}$$

For squared error, we have

$$\nabla_z^2 L(z) = 2 J(z)^\top J(z) + \sum_k \big(f_k(z) - y_k\big) \nabla_z^2 f_k(z), \tag{14}$$

with $J(z) = \partial_z f(z, \theta)$. For unbiased estimates $f_k(z)$ the residual in the second term is negligible on average, so substituting back into the expectation gives

$$\mathbb{E}_\epsilon[L(z')] \approx L(z) + \frac{\sigma^2}{2} \operatorname{Tr}\Big(2 J(z)^\top J(z) \, G^{-1}(z)\Big) \tag{15}$$

$$= L(z) + \sigma^2 \operatorname{Tr}\big(J(z)^\top G^{-1}(z) \, J(z)\big). \tag{16}$$

where the last equality uses cyclicity of the trace. The additive term is the *induced regularizer* from corrupting representations with Gaussian noise of covariance $\sigma^2 G^{-1}(z)$. It penalizes large output gradients weighted by the manifold's predefined inverse metric, aligning decoder smoothness with local curvature. We analyze its effects further in Appendix F. Our concrete implementation mirrors a single Riemannian gradient descent step, but here scaling and retracting a *noise vector* to the manifold rather than a gradient vector (details in Appendix A).

### 3.2 DATASETS

**Cell cycle stages.** Measuring gene expression of individual fibroblasts with single-cell RNA sequencing captures a continuous, asynchronous progression through the cell division cycle. Transcriptomic changes occur through these phases, yielding cyclic patterns in gene expression. As the data is not coupled in nature (we cannot identify and keep track of individual cells), unsupervised learning is suitable for picking up patterns about the underlying distribution of cells.

We apply our *Riemannian generative decoder* to the human fibroblast scRNA-seq dataset (5 367 cells × 10 789 genes) introduced in DeepCycle (Riba et al., 2022) and archived on Zenodo (Riba, 2021). Data were already preprocessed by scaling each cell to equal library size, log-transforming gene counts, and smoothing and filtering using a standard single-cell pipeline. Before modeling, we subsampled to 189 genes annotated with the cell cycle gene ontology term (GO:0007049) retrieved via QuickGO (Binns et al., 2009) in accordance with other cell cycle studies.

**Branching diffusion process.** The synthetic dataset from Mathieu et al. (2019)[1] simulates tree-structured data via a hierarchical branching diffusion: from a root at the origin in $\mathbb{R}^d$ we grow a depth-$D$ tree where each node at depth $\ell$ produces $C$ children by

$$x_{\text{child}} = x_{\text{parent}} + \epsilon, \quad \epsilon \sim \mathcal{N}\left(0, \frac{\sigma_b^2}{p^\ell} I\right). \tag{17}$$

For each node we also generate $S$ noisy sibling observations $x_{\text{obs}} = x_{\text{node}} + \epsilon'$ with $\epsilon' \sim \mathcal{N}\left(0, \frac{\sigma_b^2}{f p^\ell} I\right)$. The dataset comprises all the noisy $x_{\text{obs}}$ and is standardized to zero mean and unit variance. We set $d = 50$, $D = 7$, $C = 2$, $\sigma_b = 1$, $p = 1$, $S = 50$, $f = 8$, yielding 6 350 observations.

**Human mitochondrial DNA.** Human mitochondrial DNA (hmtDNA) is a small, maternally inherited genome found in cells' mitochondria. Its relatively compact size and stable inheritance make it a fundamental genetic marker in studies of human evolution and population structure. A relatively rapid mutation rate has led the genomes to distinct genetic variants, named *haplogroups*, which reflect evolutionary branching events.

We retrieved 67 305 complete or near-complete sequences (15 400 – 16 700 bp) from GenBank via a query from MITOMAP (MITOMAP, 2023). Sequences were annotated with haplogroup labels using *Haplogrep3* (Schönherr et al., 2023), leveraging phylogenetic trees from PhyloTree Build 17

---

[1]Available under MIT license

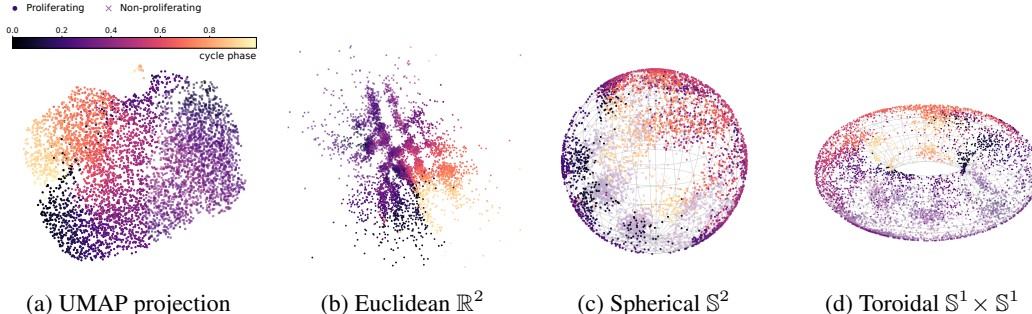

(a) UMAP projection     (b) Euclidean $\mathbb{R}^2$     (c) Spherical $\mathbb{S}^2$     (d) Toroidal $\mathbb{S}^1 \times \mathbb{S}^1$

Figure 2: **Cell cycle phases using either (a) UMAP or (b–d) different Riemannian manifolds.** Samples are concatenated across train/validation/test sets. The phase is inferred by DeepCycle as a continuous variable $\phi \in [0, 1)$ which wraps around such that $\phi = 0$ and $\lim_{\phi \to 1} \phi$ denote the same point in the cycle. Best viewed zoomed in.

Table 1: **Cell cycle: Correlation and reconstruction metrics across five random initializations** (formatted as mean $\pm$ std). Pearson/Spearman correlate phase distances to latent distances while MAE/MSE measure reconstruction by L1/L2-norm. Our models in gray. Comparisons with $\mathcal{S}$-VAE (Davidson et al., 2018) and $\Delta$VAE (Rey et al., 2019).

| | Train | | | | Test | | | |
|---|---|---|---|---|---|---|---|---|
| | Pearson | Spearman | MAE | MSE | Pearson | Spearman | MAE | MSE |
| Euclidean $\mathbb{R}^2$ | $0.47_{\pm 0.03}$ | $0.50_{\pm 0.03}$ | $0.31_{\pm 0.00}$ | $0.17_{\pm 0.00}$ | $0.52_{\pm 0.03}$ | $0.53_{\pm 0.03}$ | $0.31_{\pm 0.00}$ | $0.18_{\pm 0.00}$ |
| Euclidean $\mathbb{R}^3$ | $0.50_{\pm 0.05}$ | $0.54_{\pm 0.04}$ | $\mathbf{0.30}_{\pm 0.00}$ | $\mathbf{0.16}_{\pm 0.00}$ | $0.55_{\pm 0.04}$ | $0.57_{\pm 0.03}$ | $\mathbf{0.31}_{\pm 0.00}$ | $\mathbf{0.17}_{\pm 0.00}$ |
| Sphere $\mathbb{S}^2$ | $\mathbf{0.58}_{\pm 0.03}$ | $\mathbf{0.59}_{\pm 0.03}$ | $0.31_{\pm 0.00}$ | $0.17_{\pm 0.00}$ | $\mathbf{0.60}_{\pm 0.03}$ | $\mathbf{0.60}_{\pm 0.03}$ | $0.32_{\pm 0.00}$ | $0.18_{\pm 0.00}$ |
| Torus $\mathbb{S}^1 \times \mathbb{S}^1$ | $0.50_{\pm 0.07}$ | $0.51_{\pm 0.07}$ | $0.31_{\pm 0.00}$ | $0.17_{\pm 0.00}$ | $0.52_{\pm 0.07}$ | $0.53_{\pm 0.08}$ | $0.32_{\pm 0.00}$ | $0.18_{\pm 0.00}$ |
| $\mathcal{S}$-VAE sphere | $0.50_{\pm 0.02}$ | $0.53_{\pm 0.03}$ | $0.32_{\pm 0.00}$ | $0.19_{\pm 0.00}$ | $0.53_{\pm 0.02}$ | $0.55_{\pm 0.02}$ | $0.32_{\pm 0.00}$ | $0.19_{\pm 0.00}$ |
| $\Delta$VAE sphere | $0.52_{\pm 0.01}$ | $0.55_{\pm 0.01}$ | $0.31_{\pm 0.00}$ | $0.17_{\pm 0.00}$ | $0.57_{\pm 0.02}$ | $0.59_{\pm 0.02}$ | $0.32_{\pm 0.00}$ | $0.18_{\pm 0.00}$ |
| $\Delta$VAE torus | $0.43_{\pm 0.07}$ | $0.45_{\pm 0.07}$ | $0.31_{\pm 0.00}$ | $0.17_{\pm 0.00}$ | $0.48_{\pm 0.06}$ | $0.50_{\pm 0.07}$ | $0.32_{\pm 0.00}$ | $0.18_{\pm 0.00}$ |

(Van Oven, 2015). A sequence was kept if the reported quality was higher than $0.9$. In addition to haplogroup classification, *Haplogrep3* identified mutations with respect to a root sequence; here, separate datasets were made using either the rCRS (revised Cambridge reference sequence) or RSRS (reconstructed Sapiens reference sequence). Mutations were then encoded in a one-hot scheme, removing mutations with $\leq 0.05$ frequency, resulting in datasets with shapes $61665 \times 6298$ (rCRS) and $57385 \times 5366$ (RSRS). Appendix D displays further characteristics.

## 4 RESULTS AND DISCUSSION

In the following, we treat each dataset to evaluate and discuss applications of unsupervised learning on meaningful geometries.

### 4.1 CELL CYCLE STAGES

Figure 2 shows latent representations learned with different manifolds on the scRNA-seq data containing an underlying cyclical biological process. While we may have an idea of an explainable global optimum — e.g., a neatly arranged circle following the cell cycle stages — optimization of the neural network does not necessarily follow such an idea. Given a model expressive enough, representations lying in a circle could as well be unrolled or have distinct arcs interchanged without any loss in task accuracy. To compare model fidelity and how well manifold distances correspond to the biological geometry, Figure 1 lists reconstruction fidelities and correlations of phase distances versus manifold geodesic distances. Here we compared to $\mathcal{S}$-VAE (Davidson et al., 2018) and $\Delta$VAE (Rey et al., 2019); see Appendix C for further details. Euclidean $\mathbb{R}^3$ yields best reconstructions

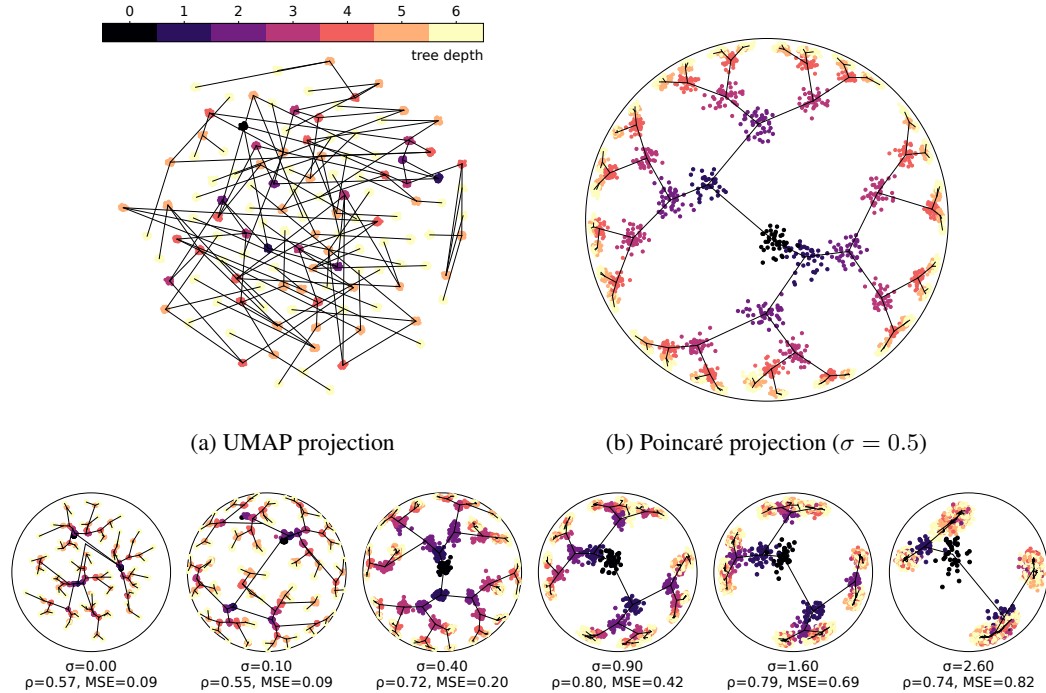

(a) UMAP projection  (b) Poincaré projection ($\sigma = 0.5$)

σ=0.00  σ=0.10  σ=0.40  σ=0.90  σ=1.60  σ=2.60
ρ=0.57, MSE=0.09  ρ=0.55, MSE=0.09  ρ=0.72, MSE=0.20  ρ=0.80, MSE=0.42  ρ=0.79, MSE=0.69  ρ=0.74, MSE=0.82

(c) Progressively increasing regularization noise of the training process (6 distinct models)

Figure 3: **Visualizations of the branching diffusion process.** Trees consist of 7 levels with color lightness denoting depth. (a) UMAP projection; (b) Poincaré disk projection of Lorentz latents using geometric regularization ($c = 5.0, \sigma = 0.5$); (c) Ablation study showing the influence of the noise scale $\sigma$, listing Pearson correlation $\rho$ and mean squared error on the training set.

Table 2: **Branching diffusion: Correlation and reconstruction metrics across five random initializations** (formatted as mean $\pm$ std). Pearson and Spearman correlate all pairs of distances in the tree structure with latent geodesic distances. Our models in gray. Comparison with $\mathcal{P}$-VAE (Mathieu et al., 2019).

| | Train | | | | Test | | | |
|---|---|---|---|---|---|---|---|---|
| | Pearson | Spearman | MAE | MSE | Pearson | Spearman | MAE | MSE |
| *Euclidean* $\mathbb{R}^2$ ($\sigma = 0.0$) | *0.53*±0.01 | *0.49*±0.01 | ***0.14***±0.00 | ***0.03***±0.00 | *0.52*±0.02 | *0.49*±0.02 | *0.18*±0.01 | *0.06*±0.01 |
| *Sphere* $\mathbb{S}^2$ ($\sigma = 0.0$) | *0.56*±0.02 | *0.53*±0.03 | ***0.14***±0.00 | ***0.03***±0.00 | *0.55*±0.02 | *0.52*±0.02 | ***0.17***±0.00 | ***0.05***±0.00 |
| Lorentz $\mathbb{H}^2$ ($\sigma = 0.1$) | 0.52±0.02 | 0.48±0.02 | 0.15±0.00 | 0.04±0.00 | 0.48±0.02 | 0.45±0.03 | 0.19±0.02 | 0.08±0.02 |
| Lorentz $\mathbb{H}^2$ ($\sigma = 0.5$) | 0.78±0.02 | 0.74±0.03 | 0.32±0.01 | 0.18±0.02 | 0.69±0.03 | 0.69±0.03 | 0.28±0.02 | 0.14±0.02 |
| Lorentz $\mathbb{H}^2$ ($\sigma = 1.0$) | **0.81**±0.02 | **0.77**±0.02 | 0.49±0.01 | 0.39±0.01 | **0.80**±0.02 | **0.76**±0.02 | 0.36±0.01 | 0.21±0.01 |
| Lorentz $\mathbb{H}^2$ ($\sigma = 2.0$) | 0.77±0.04 | 0.74±0.05 | 0.68±0.01 | 0.74±0.01 | 0.79±0.09 | 0.73±0.11 | 0.52±0.02 | 0.45±0.03 |
| $\mathcal{P}$-VAE $\mathbb{B}^2$ ($c = 1.2$) | 0.68±0.03 | 0.54±0.07 | 0.42±0.02 | 0.30±0.02 | 0.68±0.04 | 0.54±0.09 | 0.42±0.02 | 0.31±0.02 |

(having more degrees of freedom), while $\mathbb{S}^2$ improves correlation with the geometry. Toroidal embeddings show greater run-to-run variability, likely due to the limited expressivity of learning on circles $\mathbb{S}^1$ embedded in 2D. While our method significantly outperforms other models on the training data, the test results are generally similar across methods from different studies.[2]

## 4.2 BRANCHING DIFFUSION PROCESS

We find that hyperbolic spaces can efficiently be used as a tool to uncover hierarchical processes. Notably, the UMAP projection (Figure 3a) fails to reveal any underlying tree topology, despite clear

---

[2]Used as a dimensionality reduction technique, generalization performance is generally of little significance.

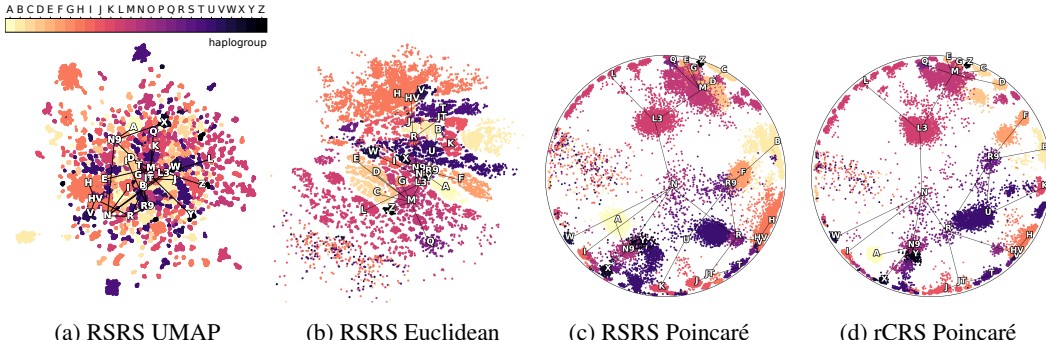

| (a) RSRS UMAP | (b) RSRS Euclidean | (c) RSRS Poincaré | (d) rCRS Poincaré |

Figure 4: **Visualizations of hmtDNA haplogroups** using either (a) UMAP, (b) Euclidean latent space, or (c–d) Poincaré projection of Lorentz latents ($c = 5.0, \sigma = 0.5$). Edges represent simplified lineage (Lott et al., 2013), nodes indicate median haplogroup positions. Best viewed zoomed in.

cluster separation. In contrast, regularized hyperbolic embeddings in the Poincaré disk (Figure 3b) recover the tree topology.

To study the effect of geometric regularization, Figure 3c shows models trained by fixing curvature $c = 5.0$ and varying noise from $\sigma = 0$ to $\sigma = 2.6$. Correlations increase sharply up to $\sigma \approx 0.9$, beyond which the noise completely overwhelms the decoder's capacity to preserve pairwise distances. This highlights a tradeoff between preserving local accuracy and enforcing global geometry. Appendix F analyzes and shows how curvature and noise level relate to each other. Appendix K tracks metric coherency as correlation between manifold versus data-space distance during training; our results show that geometric noise drives correlation steadily higher than no noise or explicit regularization. This indicates our model succeeds in internalizing the prescribed geometry as an inductive bias.

### 4.3 TRACING HUMAN MIGRATIONS

By examining differences in hmtDNA sequences, it is possible to infer patterns of migration, lineage, and ancestry. In Figure 4, we show simplified lineages based on Lott et al. (2013) and Van Oven (2015). The figure shows how UMAP fails in uncovering the hierarchical nature (panel *a*) while Euclidean embeddings show slight improvement (panel *b*). Hyperbolic models *clearly recover haplogroup hierarchies* (panels *c–d*), regardless of which reference sequence was used to encode the data. This choice otherwise has a big impact on the actual sequence encodings, preprocessing and filtering (see Appendix D), but models on either set converge at strikingly similar representations when using the same seed (panels *c–d*). The geographical locations of haplogroups strongly correspond to the locations of representations when comparing with migration maps from, e.g., Lott et al. (2013).

Appendix E lists correlation and reconstruction metrics for hmtDNA models (with one-hot data-space distance as a proxy for tree distance). For regularized hyperbolic manifolds we found mainly Spearman correlations to improve, denoting a *non-linear correlation*. Intuitively, the manifold succeeds in capturing the hierarchical branching structure of the haplogroup tree, but absolute path lengths are rescaled by the curvature. Figure 4 strongly suggests that hyperbolic distances better relate to the tree structure than Euclidean ones. In Section 4.4, we treat additional quantitative results based on hmtDNA metadata.

### 4.4 GENERAL UTILITY

We assess (i) generative fidelity with a discrimination test, (ii) downstream predictive utility from learned latents, and (iii) wall time per epoch. Since many tricks can however increase utility and generative metrics (e.g., training with a generative adversarial loss or engineering complex decoders), these evaluations act mainly as sanity checks.

**Matches or improves generative performance.** An XGBoostClassifier (default parameters) is trained to distinguish (1) optimized reconstructions of real test samples versus (2) reconstructions obtained by sampling $z \sim p(z)$ and decoding $p(x \mid z)$. We use half of the cell cycle test set to train

Table 3: **Human mitochondrial DNA: downstream utility (accuracy)** for logistic regression (LR) or XGBoostClassifier (XGB) on rCRS latents. Trends were consistent for RSRS. Our models in gray.

| Manifold | Region (3-way) | | Haplo 1 (24-way) | | Haplo 2 (128-way) | |
|---|---|---|---|---|---|---|
| | LR | XGB | LR | XGB | LR | XGB |
| Hyperbolic $\mathbb{H}^2_{\sigma=0.1}$ | 0.72 | 0.90 | 0.49 | 0.74 | 0.31 | 0.42 |
| Hyperbolic $\mathbb{H}^2_{\sigma=0.5}$ | **0.86** | **0.97** | **0.70** | **0.85** | **0.43** | 0.41 |
| Euclidean $\mathbb{R}^2$ | 0.69 | 0.85 | 0.46 | 0.74 | 0.31 | **0.43** |
| $\mathcal{P}$-VAE $\mathbb{H}^2$ | 0.52 | 0.65 | 0.19 | 0.44 | 0.13 | 0.41 |

Table 4: **Runtime (s) per epoch** on the cell cycle dataset (all genes) for varying latent dimension (including geometric noise); formatted as mean $\pm$ std through 100 epochs after warmup. Expl. RGD refers to explicit intrinsic curvature regularization (Lee & Park, 2023). *1: Breaks in computing the manifold volume from a value factorial in d. 2: Breaks to numerical instability (authors only attempt low dimensionality). 3: Computationally challenging (authors only attempt low-dimensionality).*

| Latent $d$ | Ours | | | $\Delta$VAE (Eucl.) | $\Delta$VAE (Sphere) | $\mathcal{P}$-VAE (Hyp.) | Expl. RGD (Hyp.) |
|---|---|---|---|---|---|---|---|
| | Eucl. | Sphere | Hyp. | | | | |
| **5D** | $0.21_{\pm0.01}$ | $0.23_{\pm0.02}$ | $0.25_{\pm0.00}$ | $0.41_{\pm0.02}$ | $0.60_{\pm0.06}$ | $1.70_{\pm0.11}$ | $7155_{\pm174}$[3] |
| **50D** | $0.23_{\pm0.03}$ | $0.24_{\pm0.01}$ | $0.27_{\pm0.00}$ | $0.45_{\pm0.05}$ | $0.89_{\pm0.02}$ | breaks[2] | infeasible[3] |
| **500D** | $0.26_{\pm0.01}$ | $0.33_{\pm0.01}$ | $0.39_{\pm0.01}$ | $0.70_{\pm0.03}$ | breaks[1] | breaks[2] | infeasible[3] |

the discriminator and the other half to evaluate it; an equal number of synthetic samples is drawn for both splits. Our results indicate that synthetic RGD generations are at least as hard to distinguish from real data reconstructions as generations from VAE baselines (Table S2).

**Matches or improves downstream utility.** We evaluate downstream performance by fitting both a simple model (logistic/linear regression) and a complex model (XGBClassifier/-Regressor) for classification and regression tasks on latents, respectively. On cell cycle latents, categorical cell stage and continuous cyclic phase yield near-identical scores across methods (Table S3). On hmtDNA latents, classification of geographical region (3-way), haplogroup first letter (24-way), and first two letters (128-way) strongly favor our model with a regularized hyperbolic space (Table 3).

**Unlocks scalability to higher latent dimensionality.** We probed feasibility at higher $d$ on the *full* cell cycle dataset (all genes; typical scRNA-seq analyses use $d \approx 50$). Timing results on one Intel® Xeon® Gold 6430 core with decoder layers [64, 128, 256] shows stable scaling across manifolds whereas variational baselines scale poorly and become numerically brittle (Table 4).

## CONCLUSIONS AND FUTURE DIRECTIONS

We introduced a unifying framework for representation learning on any Riemannian manifold by combining Riemannian optimization with an encoder-less generative model. This simplifies learning since we avoid density estimations, challenging for a general setting of manifolds. With a novel geometric regularization based on noise perturbation, our empirical validations demonstrated our model to successfully capture intrinsic geometric structures across diverse datasets, substantially improving correlations between latent distances and ground truth geometry. While we studied simple, low-dimensional manifolds in an explorative setting, our method unlocks higher latent dimensionality as well as heterogeneous manifold combinations, notoriously difficult with current methods. Future research directions include adaptive geometric regularization strategies, extensions to manifold-valued network weights, and exploring latent manifold structures within pretrained neural networks (e.g., generative diffusion processes or progressive generations from language models).

The decoder-only framework stores each representation explicitly, yielding memory that grows linearly with dataset size. This per-sample parameterization may be prohibitive for datasets of millions of points. Hybrid schemes — such as amortized inference or low-rank factorization — could mitigate this. Lastly, our curated hmtDNA dataset invites for further empirical studies, including analyses of geographic distances, migration patterns, or distortion-based metrics of the common consensus trees. We make the data easily available on (redacted).

## REPRODUCIBILITY STATEMENT

Anonymized code, configs, and scripts to reproduce results are provided at the repository linked in the abstract. Data preprocessing, splits, and hyperparameters are specified in Section 3, Appendix A, Appendix D, and Appendix C; ablations and settings for curvature/noise are in Appendix F.

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

## A  TRAINING DETAILS

```
model.z     := init_z(n, manifold) # initialize points on a manifold   1
model_optim := Adam(model.decoder.parameters())                        2
rep_optim   := RiemannianAdam([model.z])                               3
                                                                        4
for each epoch:                                                         5
    rep_optim.zero_grad()                                              6
    for each (i, data) in train_loader:                               7
        model_optim.zero_grad()                                       8
        z    := model.z[i]                                            9
        z    := add_noise(z, manifold, std) # optional geometric noise  10
        y    := model(z)                                             11
        loss := loss_fn(y, data)                                     12
        loss.backward()                                              13
        model_optim.step()                                           14
    rep_optim.step()                                                 15
```

Listing S1: Pseudocode for training the Riemannian generative decoder.

```
def add_noise(manifold, z, std):                                       1
    noise     := sample_normal(shape=z.shape) * std                   2
    rie_noise := manifold.egrad2rgrad(x, noise)                       3
    z_noisy   := manifold.retr(x, rie_noise)                          4
    return z_noisy                                                    5
```

Listing S2: Pseudocode for adding geometric noise. `egrad2rgrad` takes a Euclidean gradient and maps it to the Riemannian gradient in the tangent space using the inverse metric. `retr` retracts a tangent vector back onto the manifold via the exponential map if a closed form is available, otherwise a first-order approximation. *geoopt* implements both functions for a wide range of manifolds.

## B  OVERVIEW OF AVAILABLE MANIFOLDS

The following are manifolds implemented in *geoopt* (Kochurov et al., 2020), applicable for our representation learning.

- Euclidean
- Stiefel
- CanonicalStiefel
- EuclideanStiefel
- EuclideanStiefelExact
- Sphere
- SphereExact

- Stereographic
- StereographicExact
- PoincareBall
- PoincareBallExact
- SphereProjection
- SphereProjectionExact
- Scaled

- ProductManifold
- Lorentz
- SymmetricPositiveDefinite
- UpperHalf
- BoundedDomain

Parameterization and further details appear on `geoopt.readthedocs.io`.

# C EXPERIMENTAL DETAILS

## C.1 PROTOCOLS AND REPRODUCIBILITY

**General details.** Non-overlapping train/validation/test splits were made using 82/9/9 percent of samples for each distinct dataset. Across data modalities, our models all use linear layers with hidden sizes $[16, 32, 64, 128, 256]$, and sigmoid linear units (SiLU) as the non-linearity between layers. Decoder parameters were optimized via *Pytorch* (Paszke et al., 2019) with Adam (learning rate $2 \times 10^{-3}$, $\beta = (0.9, 0.995)$, weight decay $10^{-3}$), a CosineAnnealingWarmRestarts schedule ($T_0 = 40$ epochs) and early-stopped with patience 85, typically resulting in approximately 500 epochs. Representations were optimized with *geoopt*'s RiemannianAdam (learning rate $1 \times 10^{-1}$, $\beta = (0.5, 0.7)$, stabilization period 5). Spherical/toroidal representations used learning rate $4 \times 10^{-1}$ and decoder $\beta = (0.7, 0.9)$. Representations are only updated once an epoch, necessitating larger learning rates and less rigidity via the beta parameters. For hyperbolic manifolds, the curvature was fixed at $c = 5.0$ unless stated otherwise. The cell cycle and branching diffusion models used mean squared error as reconstruction objective, while the hmtDNA models used binary cross entropy.

**Initializing latents.** Initialization strategies for traditional models have been tuned for long. We however simply follow the strategy:

- *Before training:* initial guesses consist of a small degree of random noise projected to the origin of the manifold if it exists, otherwise around a random point. Randomly covering the entire manifold is generally not suitable, since latents cannot easily jump.
- *After training:* a number of initial guesses for each point are sampled from around the manifold, and we continue optimization from the one with smallest loss. If there are classes or other distinct regions on the latent manifold, sampling each region is a natural approach. This is a fast and simple variant; one may also train shortly in each sampled location before committing to any.

**Test-time RGD latents.** Following the strategy of Schuster & Krogh (2023), test-time representations for our model were found by freezing the model parameters and finding optimal $z$ through maximizing the log-likelihood of Equation 10.

**UMAP parameters.** For both the branching diffusion and hmtDNA UMAPs (Figure 3a and Figure 4a), hyperparameters `n_neighbors=30` and `min_dist=0.99` were used to help promote global structure. For Figure 2a, the UMAP coordinates of the original study were used.

**Comparison details.** We evaluated existing implementations of three baselines: the $\mathcal{P}$-VAE of Mathieu et al. (2019) (based on MIT-licensed `https://github.com/emilemathieu/pvae`), the $\mathcal{S}$-VAE of Davidson et al. (2018) (based on MIT-licensed `https://github.com/nicola-decao/s-vae-pytorch`), and the $\Delta$VAE of Rey et al. (2019) (based on Apache 2.0-licensed `https://github.com/luis-armando-perez-rey/diffusion_vae`). Model architectures were fixed — here, implementations of earlier methods were adjusted to use the same architectural backbone as ours (see the *General details* paragraph) — while hyperparameters were tuned for each model and dataset.

## C.2 HARDWARE

All experiments were carried out on a Dell PowerEdge R760 server running Linux kernel 4.18.0-553.40.1.el8_10.x86_64. Key components:

- **CPU**: $2 \times$ Intel® Xeon® Gold 6430 (32 cores/64 threads per CPU, 2.1 GHz base)
- **Memory**: 512 GiB DDR5-4800 ($8 \times 64$ GiB RDIMMs)
- **GPUs**: $1 \times$ NVIDIA A30 (24 GB HBM2e; CUDA 12.8; Driver 570.86.15)

Training single-cell and branching diffusion models takes a few minutes on our setup; models on the mitochondrial DNA data train for around 20 minutes.

## D    HMTDNA DATA DISTRIBUTIONS

[Figure S1](#) shows the distribution of mutation counts for datasets using different reference sequences.

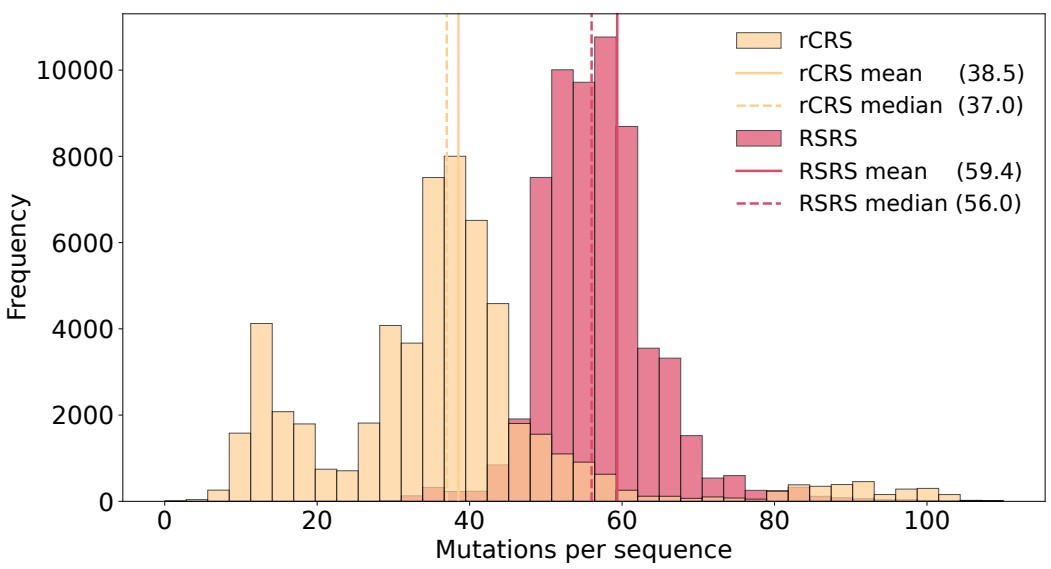

Figure S1: **Distributions of mutation counts** for datasets using different root sequence. Sequence counts of each dataset differ since the choice of haplo-tree changes the reported qualities from *Haplogrep3*, affecting the filtering procedure. Using the revised Cambridge reference sequence (rCRS) means that most sequences contain less mutations when compared to the reconstructed Sapiens reference sequence (RSRS).

## E    HMTDNA CORRELATIONS AND RECONSTRUCTIONS

In a similar fashion to the other datasets, [Table S1](#) lists correlation and reconstruction metrics for the hmtDNA dataset. It however uses one-hot data-space distance as a heuristic for tree distance.

Table S1: **Correlation and reconstruction metrics across three runs for the hmtDNA dataset** (mean $\pm$ std). Mean F1 scores assess reconstruction; Pearson and Spearman correlate manifold vs genetic distance (5000 random points). RSRS/rCRS denote distinct reference sequences.

|  | **Train** | | | **Test** | | |
|---|---|---|---|---|---|---|
|  | Pearson | Spearman | F1 | Pearson | Spearman | F1 |
| $rCRS \; \mathbb{H}^2 \; {}_{(\sigma\,=\,0.1)}$ | $0.18_{\pm0.02}$ | $0.17_{\pm0.05}$ | $0.88_{\pm0.00}$ | $-0.00_{\pm0.08}$ | $-0.04_{\pm0.13}$ | $0.74_{\pm0.01}$ |
| $rCRS \; \mathbb{H}^2 \; {}_{(\sigma\,=\,0.5)}$ | $0.28_{\pm0.01}$ | $\mathbf{0.50}_{\pm0.04}$ | $0.79_{\pm0.01}$ | $0.15_{\pm0.01}$ | $0.28_{\pm0.03}$ | $\mathbf{0.80}_{\pm0.01}$ |
| $rCRS \; \mathbb{R}^2 \; {}_{(\sigma\,=\,0.0)}$ | $\mathbf{0.41}_{\pm0.03}$ | $0.42_{\pm0.10}$ | $\mathbf{0.90}_{\pm0.00}$ | $0.16_{\pm0.07}$ | $0.24_{\pm0.14}$ | $0.73_{\pm0.02}$ |
| $RSRS \; \mathbb{H}^2 \; {}_{(\sigma\,=\,0.1)}$ | $0.15_{\pm0.01}$ | $0.12_{\pm0.04}$ | $0.93_{\pm0.00}$ | $0.04_{\pm0.10}$ | $0.04_{\pm0.23}$ | $0.83_{\pm0.02}$ |
| $RSRS \; \mathbb{H}^2 \; {}_{(\sigma\,=\,0.5)}$ | $0.28_{\pm0.01}$ | $\mathbf{0.49}_{\pm0.02}$ | $0.86_{\pm0.00}$ | $0.15_{\pm0.02}$ | $0.30_{\pm0.04}$ | $\mathbf{0.88}_{\pm0.00}$ |
| $RSRS \; \mathbb{R}^2 \; {}_{(\sigma\,=\,0.0)}$ | $\mathbf{0.35}_{\pm0.00}$ | $0.29_{\pm0.01}$ | $\mathbf{0.94}_{\pm0.00}$ | $0.12_{\pm0.07}$ | $0.23_{\pm0.09}$ | $0.83_{\pm0.02}$ |

# F  GEOMETRIC REGULARIZATION: CURVATURE VERSUS NOISE SCALE

For a hyperbolic model with curvature $c$, the metric and its inverse carry a nontrivial, state-dependent factor — which cannot be absorbed into a single global $\sigma$.

Concretely, for the ball of curvature $c$ one has

$$G(z) = \frac{4}{(1 - c\,\|z\|^2)^2}\, I, \qquad G^{-1}(z) = \frac{(1 - c\,\|z\|^2)^2}{4}\, I,$$

so the regularizer becomes

$$\mathbb{E}[L(z')] \approx L(z) + \sigma^2\, \frac{(1 - c\,\|z\|^2)^2}{4}\, \mathrm{Tr}\big(J(z)^\top J(z)\big).$$

Changing $c$ thus reshapes the weight $\frac{(1-c\,\|z\|^2)^2}{4}$ across the manifold rather than rescaling a uniform noise-variance. Only at $\|z\| \approx 0$ does it reduce to a constant factor, but in general the curvature and noise-scale contribute distinct effects.

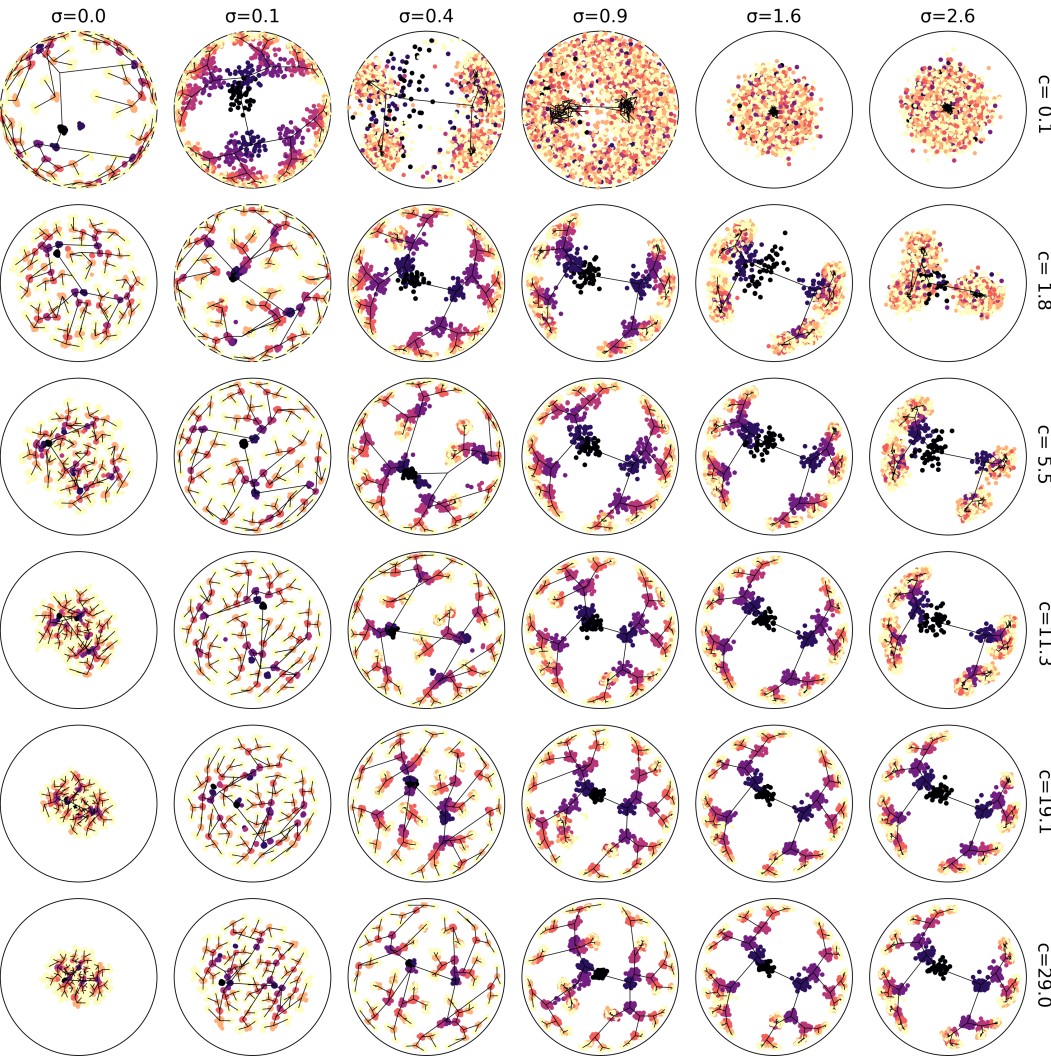

Figure S2: **Effects of manifold curvature and noise level for hyperbolic models on the synthetic branching diffusion dataset**. The visualization is similar to Figure 3c but contains a selection of curvatures rather than $c = 5.0$. Trees consist of 7 levels; color lightness denotes depth.

The local noise standard deviation induced by this Riemannian scaling is

$$\sigma(z) = \frac{\sigma\left(1 - c\,\|z\|^2\right)}{2},$$

which depends on both curvature and position. In particular,

$$\frac{\partial \sigma(z)}{\partial c} = -\frac{\sigma\,\|z\|^2}{2} < 0 \quad (\|z\| > 0),$$

so increasing $c$ *attenuates* the noise magnitude as one moves away from the origin: If you fix $\sigma$ and increase $c$, then for any $\|z\| > 0$ the factor $(1 - c\,\|z\|^2)$ is smaller, so the actual standard deviation of the injected noise at that point is reduced. Intuitively, points "away from the origin" (larger $\|z\|$) receive less noise. By contrast, raising the global noise scale $\sigma$ amplifies noise uniformly across all $z$. Thus curvature $c$ controls the spatial profile of the perturbations, whereas $\sigma$ governs their overall amplitude. Using the synthetic branching diffusion data, Figure S2 shows the effect of curvature and noise level.

## G  NOISE ABLATION ON THE HMTDNA SEQUENCES

Figure S3 shows the effect of geometry-aware regularization, now on the hmtDNA data.

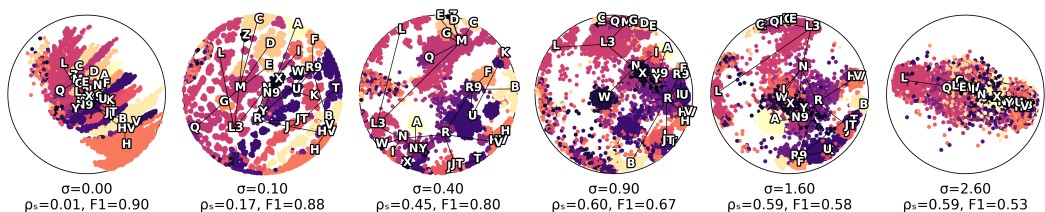

| $\sigma=0.00$ | $\sigma=0.10$ | $\sigma=0.40$ | $\sigma=0.90$ | $\sigma=1.60$ | $\sigma=2.60$ |
| $\rho_s=0.01$, F1=0.90 | $\rho_s=0.17$, F1=0.88 | $\rho_s=0.45$, F1=0.80 | $\rho_s=0.60$, F1=0.67 | $\rho_s=0.59$, F1=0.58 | $\rho_s=0.59$, F1=0.53 |

Figure S3: **Gradually increasing $\sigma$ on the rCRS hmtDNA data**, listing Spearman correlation $\rho_s$ and mean F1-score on the training set. Fixed curvature $c = 5.0$.

## H  RELATIONSHIPS OF TABLE 2

To make Table 2 more digestable, Figure S4 visualizes how the noise level $\sigma$ impacts correlation and reconstruction metrics for the synthetic branching diffusion dataset.

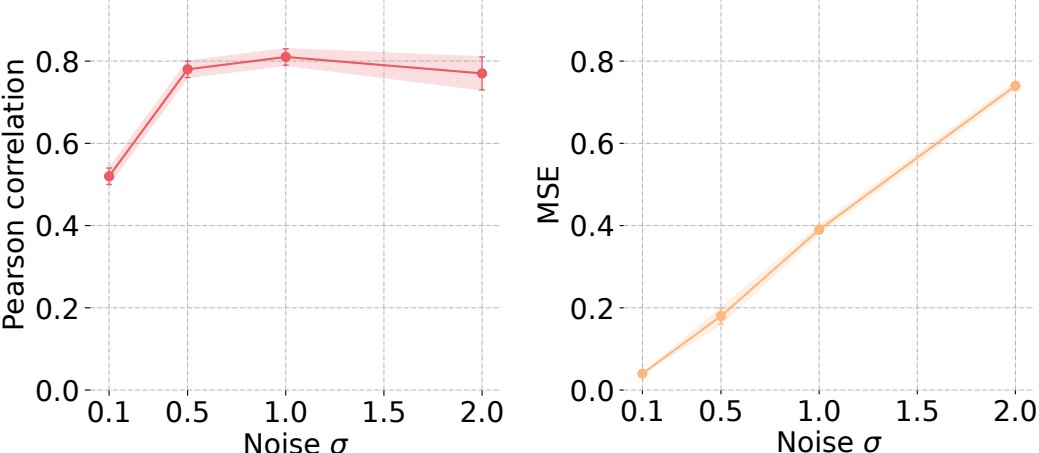

Figure S4: **Errorbar plots varying $\sigma$ for the geometry-aware regularization**; a visualization of the Lorentz results of Table 2.

## I   CELL CYCLE GENERATIVE FIDELITY

For generating scRNA-seq cells, Table S2 shows RGD generations are at least as hard to discriminate as VAE baselines.

Table S2: **Generative fidelity** measured by XGBClassifier accuracy in discriminating real reconstructions versus synthetic reconstructions (lower is better; perfectly indiscernible gives 0.5 in expectation).

| Manifold | Accuracy |
|---|---|
| RGD Sphere $\mathbb{S}^2$ | 0.58 |
| $\mathcal{S}$-VAE Sphere $\mathbb{S}^2$ | 0.58 |
| $\Delta$VAE Sphere $\mathbb{S}^2$ | 0.62 |
| RGD Torus $\mathbb{S}^1 \times \mathbb{S}^1$ | 0.59 |
| $\Delta$VAE Torus $\mathbb{S}^1 \times \mathbb{S}^1$ | 0.63 |

## J   CELL CYCLE DOWNSTREAM PERFORMANCE

Table S3 reports similar downstream performances across methods for predicting categorical cell stage and continuous cell phase from latents.

Table S3: **Cell cycle: downstream utility** — accuracy in predicting cell stage (3-way) or cyclic $R^2$ in regressing cell phase (continuous) using logistic/linear regression (LR) and XGBoost (XGB).

| | Cell stage (3-way) | | Cell phase (cont.) | |
|---|---|---|---|---|
| **Manifold** | **LR** | **XGB** | **LR** | **XGB** |
| RGD Euclidean $\mathbb{R}^2$ | 0.89 | 0.90 | 0.41 | 0.86 |
| RGD Euclidean $\mathbb{R}^3$ | 0.93 | 0.91 | 0.44 | 0.87 |
| RGD Sphere $\mathbb{S}^2$ | 0.90 | 0.89 | 0.45 | 0.87 |
| RGD Torus $\mathbb{S}^1 \times \mathbb{S}^1$ | 0.88 | 0.89 | 0.43 | 0.86 |
| $\Delta$VAE Sphere $\mathbb{S}^2$ | 0.90 | 0.90 | 0.47 | 0.88 |
| $\Delta$VAE Torus $\mathbb{S}^1 \times \mathbb{S}^1$ | 0.89 | 0.89 | 0.46 | 0.86 |
| $\mathcal{S}$-VAE $\mathbb{S}^2$ | 0.91 | 0.87 | 0.48 | 0.87 |

## K   BRANCHING DIFFUSION METRIC COHERENCY

Figure S5 shows how correlation between manifold distance versus data-space distance improves during the first 400 epochs of training hyperbolic models. Particularly our regularized model shows large positive correlation.

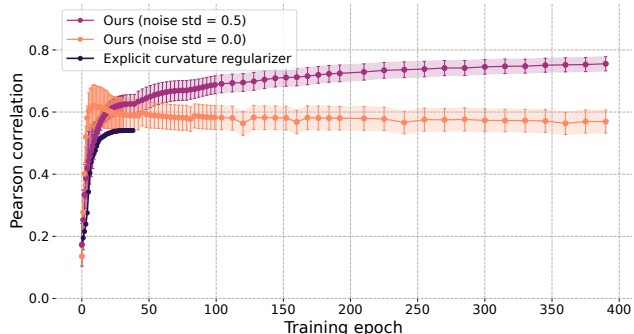

Figure S5: **Pearson correlation of manifold distance versus data-space distance during training** for hyperbolic models on the branching diffusion dataset over 5 runs. The explicit curvature model was stopped after three training days and only shows results from one run (refer to Table 4).

## L   USE OF LARGE LANGUAGE MODELS (LLMS)

LLMs were used only for language polishing. They did not contribute to model design, experiments, or analysis. All technical content was written, verified, and is the full responsibility of the authors.

