# OpenReview forum: "Riemannian generative decoder"
_ICLR.cc/2026/Conference — ICLR 2026 Conference Withdrawn Submission_

### Official Review · Reviewer_dTv6 · 2025-10-29

**Soundness:** 2
**Presentation:** 3
**Contribution:** 2
**Rating:** 4
**Confidence:** 4

**Summary:**

The authors propose an alternative method for Riemannian representation learning that does not rely on an encoder which estimates densities on chosen manifolds. Instead, they use a decoder-only framework that is trained to find manifold-valued latents for arbitrary Riemannian manifolds. The proposed method, termed Riemannian Generative Decoder, is evaluated on three datasets including both synthetic and real-world biological data to demonstrate that it can respect the prescribed geometry and capture the intrinsic non-Euclidean structure.

**Strengths:**

1. It is an interesting idea to use a decoder-only framework for Riemannian representation learning.
2. It is great that the authors considered various classes of Riemannian representation spaces (spherical, hyperbolic, general geometries) and properly used the regular versus Riemannian optimizer for different parameters.
3. Overall, the aesthetics look great. The colors and styles for most figures and tables are good.
4. Runtime comparison shows significant advantage over most encoder-decoder alternatives.

**Weaknesses:**

While the idea is interesting, the presentation of the work has several weaknesses.

1. The decoder-only architecture is not well motivated. Why not an encoder-only or encoder-decoder architecture? The authors considered encoder-decoder architectures and mentioned that they optimize numerically brittle objectives, potentially harming model training and quality, but have not supported the claim with results. To back up their claims, I would recommend the authors show "more stable training" and "faster convergence" or something along these lines. On the other hand, the authors do not seem to even consider encoder-only architectures, which can be learned in an unsupervised fashion as well, for example by enforcing similarity of distance measures between the ambient space and latent space (see GAGA [1]).
2. It seems to me that the proposed method depends on a predefined manifold prior which is not learnable, which implies that using the method requires trying out many different types of Riemannian manifolds (and various curvatures) for any new dataset. This seems like a big disadvantage compared to data-driven alternatives that do not assume the manifold geometry.
3. The baselines are a bit limited. The authors mentioned AE, VAE and DGD, but have not included AE methods in baselines. Besides, all the baselines are using predefined manifold priors, lacking data-driven alternatives such as GAGA [1] and geometric AE [2].

[1] Geometry-Aware Generative Autoencoders for Warped Riemannian Metric Learning and Generative Modeling on Data Manifolds. AISTATS 2025.

[2] Geometric Autoencoders – What You See is What You Decode. ICML 2023.

**Questions:**

See Weaknesses.

Another minor point: For Table 4, I would appreciate if the top rule is added back.

---

> ### Author Response · Authors · 2025-12-02
>
> We sincerely appreciate the reviewer for their recognition of the decoder-only idea, the presentation, the significance of the runtime comparisons, and for their constructive criticism. We treat concerns in the following.
>
> **Motivation for Architecture**
>
> >_The decoder-only architecture is not well motivated. Why not an encoder-only or encoder-decoder architecture?_
>
> Our goal is not to argue that encoder-based architectures are inferior in general, but to show that a decoder-only framework can be a _simple_, _robust_ and _powerful_ alternative.
> Encoder-decoder Riemannian VAEs require tractable manifold priors and posteriors with reparameterization, which currently exist only for a few special manifolds and are costly to scale. Empirically, our runtime table demonstrates that such models break or become infeasible at high latent dimensions, whereas RGD remains stable. The decoder-only formulation avoids all density-related complications and only requires Riemannian optimization on $\mathcal{Z}$.
>
> In general, we replace the _variational approximation_ with _exact optimization_ on the manifold, which we in Table 4 show allows us to scale to larger dimensions (and more general settings) where alternatives fail.
>
> **Baselines**
>
> >_The baselines are a bit limited. [...] lacking data-driven alternatives such as GAGA [1] and geometric AE [2]._
>
> The paper compares against other Riemannian latent-variable models, closest possible to our methodology --- and these already show that RGD offers better geometric alignment, more favorable computational properties, and more control in choice of manifold. Focusing on such baselines ensures a fair and detailed comparison of the model framework, i.e., _our_ contribution. We however appreciate the suggested works, especially the Geometric AE [1], which we plan on covering as part of the work. For additional details, refer to our reply to Reviewer 7U44.
>
> **Instability of Encoding onto Riemannian Manifolds**
>
> >_To back up their claims, I would recommend the authors show ‘more stable training’ and ‘faster convergence’ or something along these lines._
>
> The reported instability manifests in two ways:
> - The Riemannian VAEs fail at moderate dimensions (breaking while computing manifold volumes or normalizing constants), as documented in Table 4.
> - Explicit curvature regularization requires second-order derivatives and is orders of magnitude slower and practically infeasible beyond small $d$  (Figure S5, Table 4).
>
> We will sharpen the wording to refer explicitly to these observations rather than making any broad unbacked statements.
>
> **Formatting**
>
> > _For Table 4, I would appreciate if the top rule is added back._
>
> Appreciated!
>
> ---
>
> _[1] L. Reiser, G. Patrini, T. Hofmann, and M. Welling. “Geometric Autoencoders – What You See Is What You Decode.” ICML, 2023._

---

### Official Review · Reviewer_7U44 · 2025-10-30

**Soundness:** 2
**Presentation:** 2
**Contribution:** 2
**Rating:** 2
**Confidence:** 4

**Summary:**

This paper introduces the Riemannian Generative Decoder, a decoder-only framework that learns manifold representations in latent space through combining Riemannian optimization and a standard decoder network.

Instead of using an encoder, RGD directly parameterizes the latent codes and updates them via Riemannian optimization, jointly maximizing the posterior of the reconstructed points. The paper also introduces geometric regularization that adapts to the manifold's local curvature by perturbing latent variables with noise based on the inverse Riemannian metric, which penalizes large output gradients and aligns the decoder's smoothness with the manifold's geometry.
By evaluating on three datasets, including a synthetic branching diffusion process, human migrations inferred from mitochondrial DNA, and cells undergoing a cell division cycle, the paper showed that RGD outperformed Riemannian VAE methods in correlation between latent geodesic distance and meaningful biological distance in the data input space.

**Strengths:**

1. The paper is relatively well organized and structured, with an easy-to-follow background section to properly introduce the decoder-only framework. The method section is well explained and the mathematical derivation for the proposed geometric regularization is clear and well-supported.

2. I appreciate the plots and visualizations for different geometries. The table is well-made.

3. The idea of only using decoder to learn non-Euclidean representations is very interesting. The geometric regularization proposed in this paper is clever and well-motivated. It leverages the inverse Riemannian metric to "undo” the distortion in large curved regions and to ensure that the decoder is penalized evenly across the entire manifold.

4. The training algorithm seems straightforward and relatively easy to implement as it leverages the existing Riemannian optimization packages and analytically known manifolds.

**Weaknesses:**

1. Lack of motivation and support to validate the need for decoder-only representation learning. The paper claims encoder-only or VAE representation learning is numerically brittle during optimization, but provides no theoretical evidence nor sufficient empirical evidence to back this claim. They empirically compared RGD with only 2 or 1 Riemannian VAE baseline(s) in different datasets. Including more baselines, especially those encoder-only representation learning methods, would strengthen the argument for the advantages of RGD.

2. The empirical experiments are relatively weak. As mentioned above, comparing with more comprehensive baselines (across encoder-only, AE, and VAE) would lend more support to the decoder-only framework. On the qualitative side, the paper mainly compared with UMAP, include other non-linear/non-euclidean dimension reduction methods such as PHATE, Diffusionmap, etc., would be necessary to provide a fair and comprehensive qualitative comparison.

3. The geometry learned by RGD is limited to known manifolds implemented by the Riemannian optimization package geoopt, and to apply the geometric regularization, the Riemannian metric must be analytically known, contradicting the paper's claim that it “applies to any manifold”.

4. Lack of related work on encoder-only and other geometry representations learning methods, such as Geometry-Aware Autoencoder (Sun, Xingzhi, et al.), Diffusionmap(Coifman, et al.), Poincaré Embeddings(M Nickel, et al.), and PHATE(Moon KR, et al.). For example, Geometry-Aware Autoencoder could learn any latent manifold by matching the local distance in latent space with the meaningful distance in input data space.

**Questions:**

The concept of using the decoder only for manifold representation learning is intriguing, and the geometric regularization is smart. However, it lacks solid experiments and comparisons with prior methods, which makes the motivation and arguments overall less convincing. In addition, the requirement to select the geometry first and learn it rather than learning any manifold geometry through the data limits the potential scope of this framework.

To further improve the paper, I highly recommend 1) thoroughly discussing existing literature on learning non-Euclidean latent representation methods, 2) conducting more comprehensive experiments, and 3) demonstrating the advantages of RGD with more metrics than distance correlation and reconstruction error. To further prove its downstream performance, I would also recommend testing on more than the clustering task.

Last but not least, although I appreciate the visualization and aesthetics of the plots, I often need to go back and forth between the plots and texts to understand them. The paper would be clearer if the image annotations were more self-contained.

---

> ### Author Response · Authors · 2025-12-02
>
> We sincerely thank the reviewer for noting the clarity of the method, the interest of our decoder-only idea, the motivation of our geometric regularization, and for providing constructive criticism. In the following, we treat raised concerns.
>
> **Encoding onto Riemannian Manifolds**
>
> >_The paper claims encoder-only or VAE representation learning is numerically brittle during optimization, but provides no theoretical evidence nor sufficient empirical evidence to back this claim._
>
> We explicitly provide evidence in Table 4. The $\mathcal{P}$-VAE (Hyperbolic) breaks due to numerical instability at $d=50$ and $d=500$. The $\Delta$VAE (Sphere) breaks at $d=500$. Explicit curvature regularization (Expl. RGD) becomes computationally infeasible (Table 4 and Figure S5 show the very limited manageable conditions). Our method remains stable across all tested dimensions. This empirical evidence demonstrates the brittleness of variational approximations on manifolds compared to our direct optimization approach.
>
> **Comparison Baselines**
>
> >_Lack of related work on encoder-only and other geometry representations learning methods, such as Geometry-Aware Autoencoder (Sun, Xingzhi, et al.), Diffusionmap(Coifman, et al.), Poincaré Embeddings(M Nickel, et al.), and PHATE(Moon KR, et al.)._
>
> There are many works in this area, but they often treat subtly different settings. We plan to expand the _Background_ section to improve our overview by directly including more works in the vicinity, e.g., the works provided in your review. Here we address a few important distinctions:
>
> **Poincaré Embeddings [1]:** This approach requires pre-existing relational data (graphs/edges) to optimize embeddings. Our setting is in contrast unsupervised learning (modeling the data itself, $x \in \mathbb{R}^D$) rather than supervised learning over graph edges. We already included the closely related Poincaré VAE [2], the most applicable alternative.
>
> **Diffusion Maps [3] and PHATE [4]:** While these are spectral/graph-based dimensionality reduction techniques, not generative probabilistic models (as they cannot generate new samples nor estimate likelihoods), we agree that the results from such methods are also interesting to compare with, and will be experimented with.
>
> **GAGA [5]:** This method is entirely complementary to ours. The model assumes a pre-learned manifold used to inform (match) latent distances of the autoencoder. It does not seek to find the initial reduction on its own, which is the setting of our work.
> The used baselines were selected as they handle the same exact problem: unsupervised generative modeling for learning manifold-valued representations.
>
> **Downstream Utility**
>
> >_To further prove its downstream performance, I would also recommend testing on more than the clustering task._
>
> We already evaluate more than clustering:
> - Cell cycle: prediction of discrete stages and continuous phase (Table S3).
> - hmtDNA: prediction of geographic region and haplogroup labels (3-way, 24-way, 128-way) with linear models and XGBoost (Table 3), where regularized hyperbolic latents perform best.
>
> We will emphasize these results more clearly in the main text to avoid having them overshadowed.
>
> **Self-Contained Figures**
>
> >_I often need to go back and forth between the plots and texts to understand them._
>
> In a revised version, we made captions self-contained and improved placements of floats.
>
> ---
>
> _[1] M. Nickel and K. Kiela. “Poincaré Embeddings for Learning Hierarchical Representations.” NeurIPS, 2017._
>
> _[2] E. Mathieu, C. Le Lan, C. J. Maddison, R. Tomioka, and Y. W. Teh. “Continuous Hierarchical Representations with Poincaré Variational Auto-Encoders.” NeurIPS, 2019._
>
> _[3] R. R. Coifman and S. Lafon. “Diffusion Maps.” Applied and Computational Harmonic Analysis, 21(1):5–30, 2006._
>
> _[4] K. R. Moon, D. van Dijk, Z. Wang, et al. “Visualizing Structure and Transitions in High-Dimensional Biological Data.” Nature Biotechnology, 37:1482–1492, 2019._
> _[5] X. Sun et al. “Geometry-Aware Generative Autoencoders for Warped Riemannian Metric Learning and Generative Modeling on Data Manifolds.” AISTATS, 2025._

---

### Official Review · Reviewer_Tix2 · 2025-10-31

**Soundness:** 3
**Presentation:** 3
**Contribution:** 2
**Rating:** 4
**Confidence:** 3

**Summary:**

The authors propose a method to train a decoder from a structured latent to a chosen dataset, where the structure can be, in theory, any Riemannian manifold. They propose a simple MAP training objective. Empirically, they validate their method on a variety of datasets, using a host of manifolds for their prior.

**Strengths:**

- The paper is very well-presented, easy to read. The idea behind it is simple and clear; the derivation of the objective is equally simple and elegant.
- The empirical evidence is extensive, and the set of tasks is varied enough to be convincing.
- Overall, the chosen latents’ structure seems to better represent the structure of the data according to the correlation coefficients on the ranking of the distances.
- The improvements are significant on some of the datasets. In particular, I find the evidence on the downstream tasks most convincing – in terms of the metric itself, and the relative improvements.

**Weaknesses:**

- Perhaps I have missed out on it in the paper, but I have found that there is little motivation backing the need to choose *a priori* a specific Riemannian structure for the latents of the decoder.
- Since the model is encoder-free, the retrieval of latents is a more complicated task.
- The reconstruction metrics, when available, are typically *slightly* worse than the simplest method – Euclidean.
- I think that the relevance of the correlation scores is to be disputed, despite it being intuitive in some sense: it would have to be shown that respecting the distances of the original space imply better latents. It is only a supposition unless it is a requirement set for some other reason.
- The paper could probably benefit from a listing/pseudocode for inference time.
- The theory is a bit light.

**Questions:**

- Why is it useful, unless, again, it is an explicit requirement, to impose a specific Riemannian structure on the latents?
- Is it not possible to design similarly a simple (variational) encoder-decoder architecture? Of course, it would be for further work, but what is the fundamental difficulty in designing such a general VAE?
- Why are the correlation coefficients relevant whatsoever? Or could you argue for formally why it could improve anything to have those to be higher? One of the reasons to have latents is dimensionality reduction, and therefore no isometry can be expected in lower dimensions anyway, while “good quality” latents can exist. Moreover, as shown in appendix H, the reconstruction error seems to degrade when correlation seems to augment, at least with respect to the noise regularisation.

Overall, I think it is a good paper, but that just lacks a bit of substance. I would be happy to revise my score if the authors could motivate a bit more why the Riemannian structure is relevant for the latents, especially as they are not available easily without an encoder. It seems to me that letting the optimisation procedure converge to the “best” (well, tautologically, the ones that minimise the loss) latents on its own is the best thing to do. I could be convinced otherwise in some specific settings.

---

> ### Author Response · Authors · 2025-12-02
>
> We thank the reviewer for appreciating our efforts in making the paper read well, for noting the elegance of the idea and derivation, and for providing detailed considerations. In the following, we treat the concerns briefly.
>
> **Motivation for Inductive Bias**
>
> >_I have found that there is little motivation backing the need to choose a priori a specific Riemannian structure for the latents of the decoder._
>
> The motivation is **hypothesis-driven exploration**. In scientific discovery, practitioners often possess strong prior knowledge about the underlying topology of a process (e.g., cell cycles are closed loops $\approx \mathbb{S}^1$, evolutionary trees are hyperbolic $\approx \mathbb{H}^d$). By imposing a specific geometry, we test whether the data supports that hypothesis. This will be written clearly. On the other extreme, unconstrained models often fail to capture these global topologies, as shown in Figure 2 (Cell Cycle), Figure 3 (Branching Diffusions) and Figure 4 (Phylogenetics).
>
> **On Euclidean Reconstructions**
>
> >_The reconstruction metrics, when available, are typically slightly worse than the simplest method – Euclidean._
>
> This is expected behavior. A Euclidean space $\mathbb{R}^d$ with sufficient dimensionality has infinite volume and zero curvature, allowing the model to "cheat" by overfitting noise to minimize reconstruction error (MSE) — have in mind that optimization during inference is also needed. A compact or curved manifold instead acts as a strict regularizer. A slightly higher MSE coupled with significantly higher geodesic correlation (Table 1, Table 2) indicates the model is learning the true structure rather than overfitting.
>
> **Correlation Metrics**
>
> >_Why are the correlation coefficients relevant whatsoever?_
>
> We use correlation between latent geodesic distances and biologically meaningful distances because in our tasks the “ground truth” geometry is known:
> - tree distances (branching diffusion),
> - cyclic phase (cell cycle),
> - genetically defined haplogroup relationships (hmtDNA).
>
> We do not claim that high correlation is universally optimal. Rather, in these settings higher Spearman/Pearson correlation is a direct measure that the learned latent space respects the known structure. This is precisely what practitioners want when using low-dimensional embeddings for hypothesis generation.
>
> The noise ablation (Figure 3c and Appendix G) shows an explicit trade-off: increasing $\sigma$ improves geometric alignment at the expense of some reconstruction error. This is expected for any regularizer that enforces non-Euclidean geometry in a low-dimensional latent. Importantly, downstream tasks on hmtDNA (Table 3) show that hyperbolic latents are competitive or superior for prediction despite slightly higher reconstruction MSE.
>
> **Inference Implementation**
>
> >_The paper could probably benefit from a listing/pseudocode for inference time._
>
> Agreed. Inference generally follows the exact same procedure as training, simply with decoder weights frozen.

---

### Official Review · Reviewer_JbAF · 2025-11-09

**Soundness:** 3
**Presentation:** 3
**Contribution:** 2
**Rating:** 4
**Confidence:** 2

**Summary:**

This paper proposes a new approach for learning latent representations directly on Riemannian manifolds without relying on an encoder network. The method involves training a decoder while simultaneously learning manifold-valued latent variables, thereby simplifying manifold constraints and enabling scaling to higher-dimensional and diverse manifolds. Authors introduce a geometry-aware regularization technique based on noise perturbations, which ensures that the smoothness of the decoder aligns with the curvature of the manifold, preserving the intrinsic non-Euclidean geometry of the data. The approach is validated on various datasets, including synthetic branching diffusion processes, evolutionary mitochondrial DNA data, and cyclic gene expression profiles. Overall, this provides a novel framework for representation learning on complex geometric spaces.

**Strengths:**

- The decoder-only approach for learning latent representations is simple, elegant, and generalizable to any Riemannian manifold. It avoids the complexities of density estimation.
- Empirical validation is comprehensive, including diverse real-world biological datasets that effectively capture hierarchical and cyclic structures.
- The geometry-aware regularization based on noise perturbations is insightful.

**Weaknesses:**

- The paper builds incrementally on DGD framework by Schuster & Krogh (2023), with somewhat limited novelty in methodology.
- Inference requires optimization at test time to compute latent codes, which could slow down inference.
- The tuning of regularization parameters lacks a thorough discussion and may need adaptive or automated strategies.
- There may be scalability concerns regarding memory and computation for very large datasets?

**Questions:**

Please refer to my comments.

---

> ### Author Response · Authors · 2025-12-02
>
> We thank the reviewer for the positive assessment of soundness/presentation, for noting the elegance of a decoder-only approach, and for constructive criticism in general. In the following, we treat noted concerns.
>
> **Our Contribution**
>
> >_The paper builds incrementally on DGD framework by Schuster & Krogh (2023), with somewhat limited novelty in methodology._
>
> We disagree that the contribution is merely incremental. While we utilize the DGD optimization scheme, our primary contribution is the **unified framework for representation learning on a multitude of manifolds**, circumventing the non-trivial issue of defining variational densities on complex geometries. Furthermore, we introduce a novel **geometric regularization** derived analytically from the decoder's sensitivity to the manifold metric. Neither of these methods are present in existing work. As detailed in Section 3.1, we derive:
> $$\mathbb{E}[L(z')] \approx L(z) + \sigma^2 \operatorname{Tr}(J(z)^\top G^{-1}(z) J(z))$$
> This aligns the decoder's Jacobian with the local curvature, a theoretically grounded contribution distinct from previous work.
>
> **Inference Speed**
>
> >_Inference requires optimization at test time to compute latent codes, which could slow down inference._
>
> This is an inherent design choice for exact optimization on manifolds. Amortized inference (encoders) introduces an approximation gap (the amortization gap). In scientific domains like computational biology (Section 4.3), the exactness of the representation often outweighs the computational cost of test-time optimization.
>
> **Scalability**
>
> >_There may be scalability concerns regarding memory and computation for very large datasets?_
>
> The memory complexity is linear in $N$ (dataset size). For the specific scientific applications targeted (scRNA-seq, phylogenetics), datasets typically range from $10^4$ to $10^6$ samples, which fits comfortably in memory as demonstrated in our experiments. As a relevant aside, Table 4 shows our method easily scales latent dimensionality (e.g., $d=500$) whereas competing approaches break due to numerical instabilities in the manifold-density computations.

---

### Note · Authors · 2025-12-02

**Comment:**

**Authors' Note**

We thank all reviewers for their careful assessment of our submission and for highlighting the clarity of the presentation, the elegance of a decoder-only formulation, and the empirical breadth. We also highly appreciate all the constructive questions raised. After consideration, we plan for journal submission.

Our central goal is to demonstrate that decoder-only Riemannian representation learning provides a unified, scalable alternative to variational methods on manifolds. Our method replaces variational density modeling --- fundamentally limited to a handful of tractable manifolds --- with exact Riemannian optimization combined with an analytically derived geometric regularizer. This yields a framework applicable to a broad class of manifolds, applicable where current alternatives fail. The choice of manifold is an intended inductive bias: in scientific settings, practitioners often possess explicit knowledge of the topology underlying their system, and our framework allows them to test such hypotheses directly.

Several concerns focused on baselines and scope. Our comparisons targeted the relevant methodological class --- generative models with manifold-valued latents --- and showed advantages in geometric fidelity, downstream utility, and computational scaling. Methods such as Poincaré embeddings [1] and GAGA [2] among others operate under different assumptions (relational supervision and pre-learned manifolds) and are therefore not direct comparators to the setting of our article. These perspectives are nonetheless valuable, and a future revision will contextualize better to avoid confusion.

We appreciate the reviewers’ time and constructive feedback. This we cannot stress enough. We will refer to the remarks while preparing a version with clearer motivation and related work.

---

_[1] M. Nickel and K. Kiela. “Poincaré Embeddings for Learning Hierarchical Representations.” NeurIPS, 2017._

_[2] X. Sun, et al. “Geometry-Aware Generative Autoencoders for Warped Riemannian Metric Learning and Generative Modeling on Data Manifolds.” AISTATS, 2025._

**Withdrawal Confirmation:**

I have read and agree with the venue's withdrawal policy on behalf of myself and my co-authors.